# Simulation of Saline Water Irrigation for Seed Maize in Arid Northwest China Based on SWAP Model

**Chengfu Yuan [1,2], Shaoyuan Feng [1,3,*], Zailin Huo [3] and Quanyi Ji [1]**

1   College of Hydraulic, Energy and Power Engineering, Yangzhou University, Yangzhou 225009, China
2   Jiangxi Water Resources Institute, Nanchang 330013, China
3   Centre for Agricultural Water Research in China, China Agricultural University, Beijing 100083, China
*   Correspondence: syfeng@yzu.edu.cn; Tel.: +86-514-87969205

**Abstract:** Water resource shortages restrict the economic and societal development of China's arid northwest. Drawing on groundwater resources for irrigation, field experiments growing seed maize (*Zea mays* L.) were conducted in 2013 and 2014 in the region's Shiyang River Basin. The Soil–Water–Atmosphere–Plant (SWAP) model simulated soil water content, salinity, and water–salt transport, along with seed maize yield, in close agreement with measured values after calibration and validation. The model could accordingly serve to simulate different saline water irrigation scenarios for maize production in the study area. Waters with a salinity exceeding 6.0 mg/cm$^3$ were not suitable for irrigation, whereas those between 3.0 and 5.0 mg/cm$^3$ could be acceptable over a short period of time. Brackish water (0.71–2.0 mg/cm$^3$) could be used with few restrictions. Long-term (five years) simulation of irrigation with saline water (3.0–5.0 mg/cm$^3$) showed soil salinity to increase by over 9.5 mg/cm$^3$ compared to initial levels, while seed maize yield declined by 25.0% compared with irrigation with brackish water (0.71 mg/cm$^3$). An irrigation water salinity of 3.0–5.0 mg/cm$^3$ was, therefore, not suitable for long-term irrigation in the study area. This study addressed significance issues related to saline water irrigation and serves as a guide for future agricultural production practices.

**Keywords:** Shiyang River Basin; saline water irrigation; SWAP model; soil water–salt transport; seed maize

## 1. Introduction

The shortage of water resources is a prominent problem restraining social and economic development in China's arid northwest. Accounting for 24.5% of the nation's land area, northwest China's arid regions span about $2.5 \times 10^6$ km$^2$. The total quantity of water resources in China's arid northwest is estimated at 197.9 billion m$^3$, only 5.84% of the nation's overall supply [1]. In the region, the mean quantities of water resources per capita and per hectare are 68% and 27%, respectively, of the national average [2]. The region's scarcity of precipitation and strong evaporative loss result in the extreme scarcity of surface water resources; however, shallow saline groundwater resources offer large reserves, with a salinity of 2–5 g/L amounting to 88.6 billion m$^3$ [3]. The irrigation area is about 3 million hectares in the arid area of northwest China [4]. In arid areas where freshwater resources are scarce, the exploitation of underground water resources and the use of saline water irrigation technologies in agricultural production are among the important measures implemented to solve water shortage issues. With a century-old history in foreign countries, saline water irrigation was implemented successfully on a number of occasions. Several studies undertaken on the north China plain and in China's northwest and coastal regions reported on good results with saline water irrigation practices. Studies indicated that saline water irrigation can alleviate soil drought, and the yield of some salt-tolerant

crops under saline water irrigation is similar to that under freshwater irrigation, increasing production compared with no irrigation [5,6]. However, should saline water be used improperly, it can lead to the accumulation of soil salinity, as well as changes in soil physical and chemical properties, causing soil secondary salinization and threatening crop growth [7]. How to use saline water resources to irrigate scientifically, rationally, safely, and effectively was always a key concern to researchers. Most studies on saline water irrigation employed indoor or fixed field experiments to analyze the relatively short-term influence of saline water irrigation on the soil environment and crop growth [8–11]. However, the effects of saline water irrigation represent a long-term process. When employing conventional long-term field experiments, these can be affected by many external factors, in addition to placing high demands on manpower, finances and experimental facilities. Accordingly, simulating the effects of long-term saline water irrigation on the soil environment and crop growth using mathematical models was proven an effective and economical research method. The Soil–Water–Atmosphere–Plant (SWAP) model, based on the Richards and advection dispersion equations, being a comprehensive model for water, heat, and solute transport [12], can be used to simulate the water, heat, and solute fluxes in the SPAC (Soil-Plant-Atmosphere Continuum), as well as the growth process of crops. The SWAP model was widely applied to simulate soil water–salt transport and its short and long-term impact on crops and the environment around the world [13–16]. Yang et al. [17] simulated the effects of brackish water irrigation under different irrigation quotas on soil salt accumulation and spring wheat yield in China's Hetao irrigation area. They also predicted soil root layer salt distribution and balance under long-term brackish water irrigation. Using SWAP, Kumar et al. [18] simulated crop root zone soil salt dynamics and relative wheat yield under different saline water irrigation conditions for the region of New Dehli, India. Studying spring wheat grown under a deficit of irrigation with saline water in China's Shiyang River basin, Jiang et al. [19] used SWAP to simulate and predict soil water–salt transport under different conditions. The salinization process led to the soils reaching equilibrium after a few years' use of saline water for irrigation. The SWAP model could be used to simulate the soil salt dynamics and wheat yield with acceptable accuracy under irrigated saline environment.

Northwest China's unique geographical and climatic conditions, along with its abundant solar and thermal resources, led the region's seed maize (hereafter simply maize) planting area to rise sharply and the crop to become economically important. However, few studies used the SWAP model to simulate and predict the effects of long-term saline water irrigation on soil water–salt transport and maize growth in China's arid northwest.

The objectives of this study were (1) to calibrate and validate the applicability of SWAP model by comparing simulated results with measured data drawn from field experiments; (2) to simulate and predict the long-term effects of saline water irrigation with waters of different salinity on soil water–salt transport and maize yield.

## 2. Materials and Methods

### 2.1. Field Experiments

The field experiments were conducted in 2013 and 2014 at the Shiyang River Experimental Station of China Agricultural University (37°52′ north (N) latitude, 102°52′ east (E) longitude, elevation 1581 m). The station is located on the upper–middle reaches of the Shiyang River, near Wuwei city, Gansu Province, China (Figure 1). The region has a typical continental temperate dry climate, with an annual average (1951–2013) precipitation of 164.5 mm and an annual average evaporation of 2000 mm [20]. The mean underground water table exceeds 40 m in depth below the ground surface.

Nine non-weighing lysimeters (Figure 2) were used for the experiments, each with an area of 6.66 m$^2$ (3.33 m × 2 m) and a depth of 3 m. Adjacent lysimeters were separated by concrete to avoid lateral seepage and surface runoff transfer. The bottom of each lysimeter was a cement floor. The physical and chemical properties of the soil prior to the onset of the experiments are presented in Table 1.

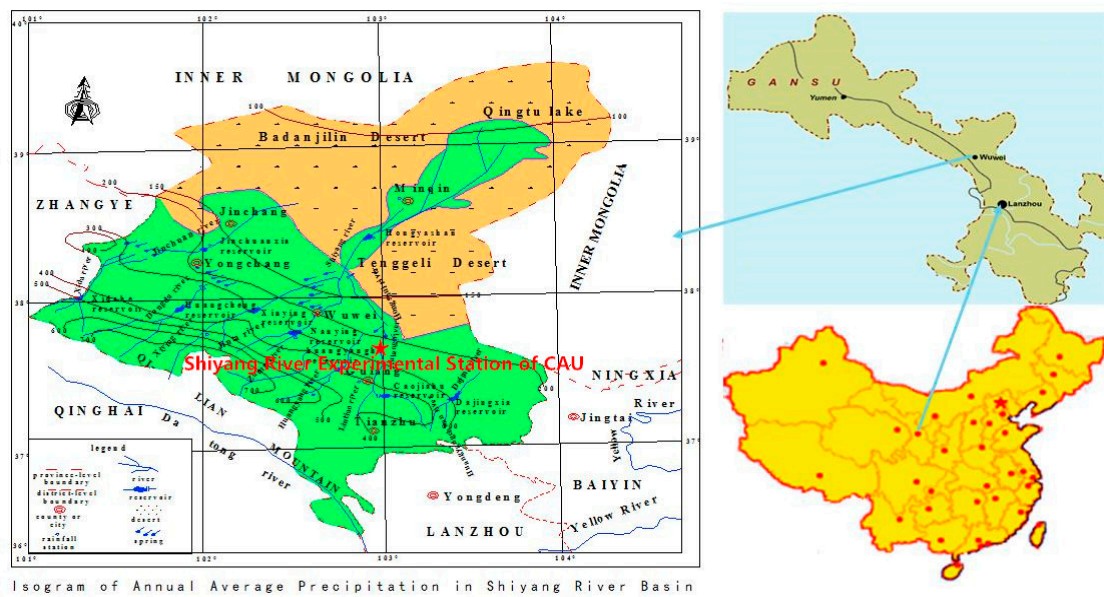

**Figure 1.** Geographical localization map of study area.

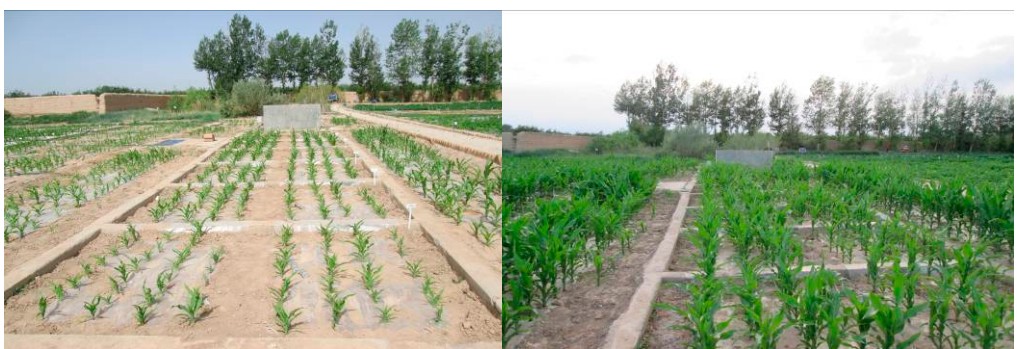

**Figure 2.** Non-weighing lysimeters used in the experiments.

**Table 1.** Soil physical and chemical properties.

| Soil Depth (cm) | Sand (%) | Silt (%) | Clay (%) | Organic Content (g/kg) | Soil bulk Density (g/cm$^3$) | Field Capacity (cm$^3$/cm$^3$) | Saturated Water Content (cm$^3$/cm$^3$) | International Soil Texture |
|---|---|---|---|---|---|---|---|---|
| 0–20 | 61.03 | 28.43 | 10.54 | 2.60 | 1.49 | 0.27 | 0.36 | Sandy loam |
| 20–40 | 58.33 | 30.45 | 11.22 | 2.64 | 1.54 | 0.30 | 0.40 | Sandy loam |
| 40–100 | 53.11 | 34.14 | 12.75 | 5.69 | 1.55 | 0.32 | 0.42 | Clay loam |

Saline water irrigation was performed with water bearing 0.71 mg/cm$^3$ (freshwater, s0), 3.0 mg/cm$^3$ (brackish water, s3), or 6.0 mg/cm$^3$ (saline water, s6). These three treatments were replicated three times, with nine test plots laid out in a split-plot arrangement (Figure 3). Freshwater bearing 0.71 mg/cm$^3$ was pumped directly from the experimental station well. Saline water bearing 3.0 mg/cm$^3$ or 6.0 mg/cm$^3$ was prepared artificially by dissolving NaCl, mgSO$_4$, and CaSO$_4$ in freshwater in mass ratio of 2:2:1. Plastic pipes equipped with water meters allowed the quantity of irrigation water to be controlled and monitored in each test pit. Maize variety "Funong 340" was sown with one line of male plants and seven lines of female plants, with 56 plants in each pit. Intra-row plant spacing was 0.25 m and inter-row spacing was 0.35 m. Sown on 20 April 2013 and 19 April 2014, maize was harvested on 13 September 2013 and 19 September 2014. The irrigation water quota refers to actual local conditions pertinent to maize production and irrigation scheduling (Table 2). Crop and field management measures followed local methods.

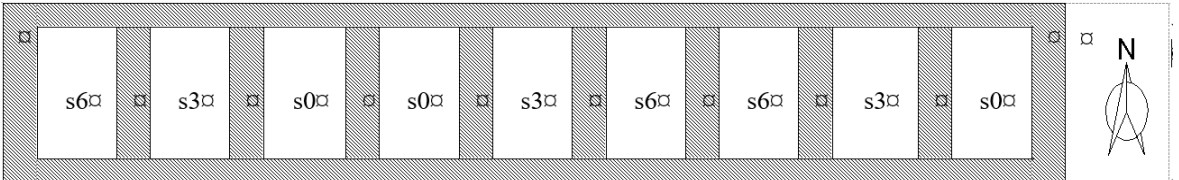

**Figure 3.** Arrangement diagram of test plot.

**Table 2.** Irrigation scheduling in the experiments.

| Treatment | Irrigation Water Salinity (mg/cm$^3$) | Irrigation Water Quota (mm) | | | | | Overall Irrigation Water Quota (mm) |
|---|---|---|---|---|---|---|---|
| | | 5 June | 30 June | 20 July | 10 August | 29 August | |
| s0 | 0.71 | 120 | 120 | 105 | 105 | 105 | 555 |
| s3 | 3.0 | 120 | 120 | 105 | 105 | 105 | 555 |
| s6 | 6.0 | 120 | 120 | 105 | 105 | 105 | 555 |

Using a soil auger, soil samples were taken prior to seeding and after harvest, and before and after each irrigation during the maize growth period in 2013 and 2014. Samples were obtained at depths of 0–10, 10–20, 20–40, 40–60, 60–80, and 80–100 cm. The soil moisture was determined gravimetrically by oven-drying soil at 105 °C for eight hours followed by weighing of the dry soil. Soil extractions were prepared at a soil-to-water ratio of 1:5. Electrical conductivity $EC_{1:5}$ was measured using an SG-3 conductivity meter (SG3-ELK742) and translated into soil salinity through the equation, *S = 0.0275EC$_{1:5}$ + 0.1366* [21]. Maize plant height, leaf length (*L*), and leaf width (*W*) were measured every 15–20 days after seedling. The *LAI* (leaf area index) was obtained through the equation *LAI = (K × L × W)/A*, where *K* is a fitting coefficient (0.75 for maize) [22], and *A* is the area covered by plant leaves. The root length density was obtained through a drilling method at different maize growth stages, followed by scanning and analysis with a root software (WinRHIZO PRO 2007). The saturated hydraulic conductivity was measured using a soil permeability meter (TST-55, China) equipped with a constant head permeameter. The soil water retention curve was measured using a high-speed refrigerated centrifuge. Soil hydraulic parameters of the VG (van Genuchten) model were analyzed through the RETC (RETention Curve) program and are presented in Table 3. The maize yield of each plot was obtained by weighting method to determine yield per hectare. The meteorological data were obtained by an automatic meteorological station (Weather Hark, Campbell Scientific, Logan, UT, USA) at the experimental station. The daily rainfall, minimum temperature, and maximum temperature for the maize growth stage are shown Figure 4. Total effective precipitation during the maize growth stage in 2013 and 2014 was 64.6 mm and 141.2 mm, respectively.

**Table 3.** Soil hydraulic parameters of VG (van Genuchten) model.

| Soil Depth (cm) | Residual Water Content (cm$^3$/cm$^3$) | Saturated Water Content (cm$^3$/cm$^3$) | Saturated Hydraulic Conductivity (cm/d) | Shape Factor $\alpha$ (/cm) | Shape Factor n | Shape Factor $\gamma$ |
|---|---|---|---|---|---|---|
| 0–20 | 0.044 | 0.36 | 32.57 | 0.024 | 1.434 | 0.5 |
| 20–40 | 0.043 | 0.38 | 29.85 | 0.024 | 1.417 | 0.5 |
| 40–100 | 0.049 | 0.40 | 13.71 | 0.011 | 1.480 | 0.5 |

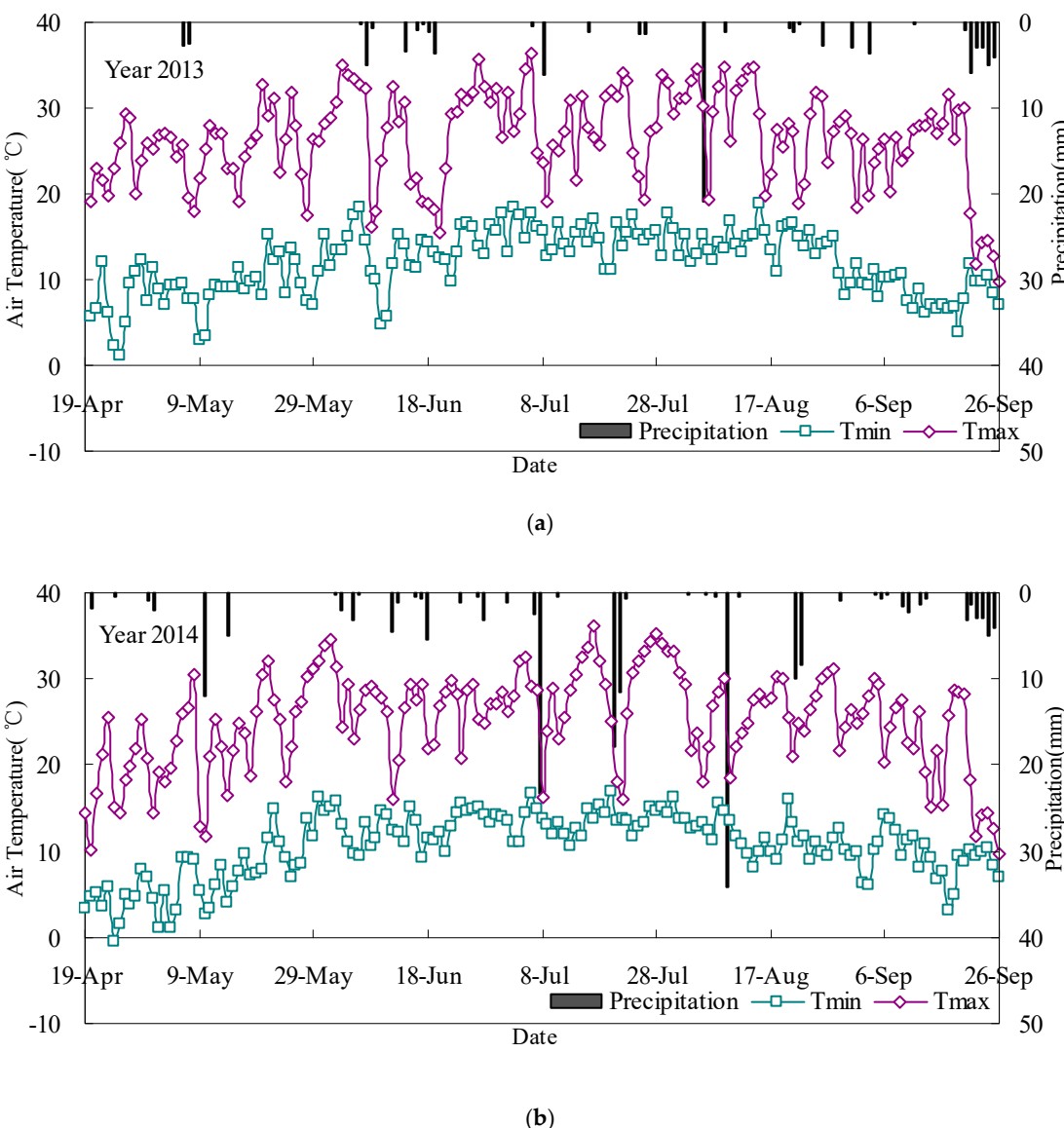

**Figure 4.** Daily meteorological data for maize growth stage in 2013 (**a**) and 2014 (**b**).

## *2.2. SWAP Model*

The SWAP model, developed by the Water Resource Group of Wageningen University, was used to simulate one-dimensional vertical soil water flow, solute transport, and crop growth [12]. The model was widely applied to simulate soil water–salt transport, crop evapotranspiration, crop growth, farmland drainage, and ground water table changing in semi-arid and arid areas around the world. The vertical soil water flow in saturated and unsaturated zone is described by Richards' equation as follows:

$$\frac{\partial \theta}{\partial t} = C(h)\frac{\partial h}{\partial t} = \frac{\partial}{\partial z}\left[K(h)(\frac{\partial h}{\partial z} + 1)\right] - S(h), \tag{1}$$

where $\theta$ is soil water content (cm$^3$/cm$^3$), $t$ is time (days), $C$ is differential water capacity(/cm), $h$ is soil water pressure head (cm), $z$ is the vertical coordinate (cm, positive uptake), $k(h)$ is the hydraulic conductivity (cm/day), and $S(h)$ is the soil water extraction rate by plant roots (cm$^3$/(cm$^3$/day)).

Solute transport is computed by the advection dispersion equation as follows:

$$J = qc - \theta(D_{dif} + D_{dis})\frac{\partial c}{\partial z}, \tag{2}$$

where $J$ is total solute flux density (g/(cm$^2$/day)), $q$ is vertical flow at the bottom (cm/day), $c$ is solute concentration in soil water (g/cm$^3$), $D_{dif}$ and $D_{dis}$ are the diffusion coefficient (cm$^2$/d) and the dispersion coefficient (cm$^2$/d), respectively, and $\partial c/\partial z$ is solute concentration gradient.

The SWAP model simulates crop growth process using the WOFOST (world food study) crop growth model, which includes a detailed crop model and a simple crop model. The simple crop growth module of Doorenbos and Kassam [23] included in SWAP was used in this study. Each growing stage, the actual yield relative to the potential yield during this growing stage is calculated by

$$1 - \frac{Y_{a,k}}{Y_{p,k}} = K_{y,k}\left(1 - \frac{T_{a,k}}{T_{p,k}}\right), \tag{3}$$

where $Y_{a,k}$ and $Y_{p,k}$ are the actual yield (kg/hm$^2$) and the potential yield (kg/hm$^2$), respectively, $T_{a,k}$ and $T_{p,k}$ are the actual transpiration (cm) and potential transpiration (cm), respectively, $K_{y,k}$ is the yield response factor of growing stage $k$, and index $k$ is the growing stage.

The relative yield of the full growing season is a product of the relative yield of each growing stage and is calculated as follows:

$$\frac{Y_a}{Y_p} = \sum_{k=1}^{n}\left(\frac{Y_{a,k}}{Y_{p,k}}\right), \tag{4}$$

where $Y_a$ and $Y_p$ are the cumulative actual yield (kg/hm$^2$) and the cumulative potential yield (kg/hm$^2$) of the entire growing season, respectively, and $n$ is the number of defined crop growing stages.

The input parameters required by the SWAP model are meteorological data, irrigation data, soil physical and chemical properties, soil hydraulic parameters, initial pressure head, salt data, crop data, and initial and boundary conditions. These parameters were obtained in the experiments. The measured dispersion length and molecular diffusion coefficient were 10 cm and 0.5 cm$^2$/day, respectively. The relative uptake of solutes by roots and the solute concentration in precipitation were both zero. The soil water pressure head and the soil salinity before sowing were regarded as initial conditions. The bottom boundary condition was set as free drainage because of the large groundwater depth. A detailed introduction to the SWAP model can be found in the SWAP model theory book [12].

The RMSE (root mean square error) and MRE (mean relative error) were used to quantify the deviation of the simulated and observed data. The RMSE and MRE were calculated according to the following equations:

$$RMSE = \sqrt{\frac{1}{N}\sum_{i=1}^{N}(P_i - O_i)^2}, \tag{5}$$

$$MRE = \frac{1}{N}\sum_{i=1}^{N}\left|\frac{P_i - O_i}{O_i}\right| \times 100\%, \tag{6}$$

where $N$ is the total number of observations in the experiments, and $P_i$ and $O_i$ are the $i$th model predicted and observed values ($i = 1, 2, \ldots, N$), respectively.

## 3. Results and Discussion

### 3.1. Calibration and Validation of SWAP Model

The model was calibrated and validated with field experiment data. The soil water content at depths of 10, 40, and 80 cm of s3 treatment in 2013 served to calibrate the SWAP model, while the soil water content of the same soil layers of s3 treatment in 2014 served to validate the model. The simulated soil water content agreed reasonably well with the measured values at different soil layers (Figure 5). The simulated values effectively reflected tendencies in variation of the measured data. The RMSE and MRE values were less than 0.05 cm$^3$/cm$^3$ and 20% in both model calibration and validation. These

model accuracy statistics showed the model performance for soil water content simulation to be good. The soil hydraulic parameters after calibration are shown in Table 4.

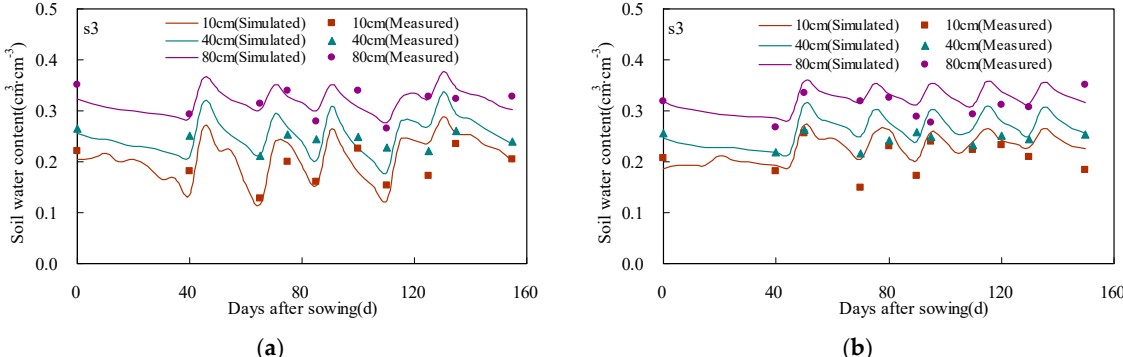

**Figure 5.** Comparison of the simulated and measured soil water content at different soil layers in calibration 2013 (**a**) and validation 2014 (**b**).

**Table 4.** Soil hydraulic parameters for different soil layers after calibration.

| Soil Depth (cm) | Residual Water Content (cm³/cm³) | Saturated Water Content (cm³/cm³) | Saturated Hydraulic Conductivity (cm/day) | Shape Factor $\alpha$ (/cm) | Shape Factor n | Shape Factor $\gamma$ |
|---|---|---|---|---|---|---|
| 0–20 | 0.055 | 0.34 | 30.81 | 0.024 | 1.402 | 0.5 |
| 20–40 | 0.044 | 0.38 | 31.66 | 0.022 | 1.408 | 0.5 |
| 40–100 | 0.095 | 0.39 | 12.00 | 0.020 | 1.310 | 0.5 |

The soil salinity at different times during the s3 treatment in 2013 served to calibrate the model, while the equivalent 2014 data served to validate the model. The simulated soil salinity agreed reasonably well with the measured data and reflected the trends in variation of the measured values at different times (Figure 6). The simulated values were mainly in especially good agreement with the measured values at the early growth stages of maize, worsening slightly at later growth stages. The RMSE and MRE values were less than 5.0 mg/cm$^3$ and 25% in both model calibration and validation. These model accuracy statistics showed the model performance for soil salinity simulation to be good. The dispersion length and molecular diffusion coefficient after calibration were 8.5 cm and 0.5 cm$^2$/day, respectively.

The yield of maize in 2013 and 2014 served, respectively, to calibrate and validate the SWAP model. Default crop parameters were adjusted within the range suggested by SWAP. The calibrated parameters were consistent with values in other maize studies [24]. The SWAP model's output was the yield relative to the maximum actual yield (6303.36 kg/hm$^2$) achieved for the s0 treatment in 2013 (set at 1.0); accordingly, the actual yield in different treatments was calculated by conversion of the simulated value. Simulated maize yield agreed reasonably well with the measured yield for the different treatments (Figure 7). The simulated maize yield mainly reflected the trends in measured yield. The RMSE and MRE values were less than 380 kg/hm$^2$ and 10% in both model calibration and validation. Model performance for maize yield simulation was good. The minimum canopy resistance, critical level $EC_{max}$, and decline per unit $EC_{slope}$ of maize after calibration were 60 s/m, 1.7 dS/m, and 4.0%, respectively.

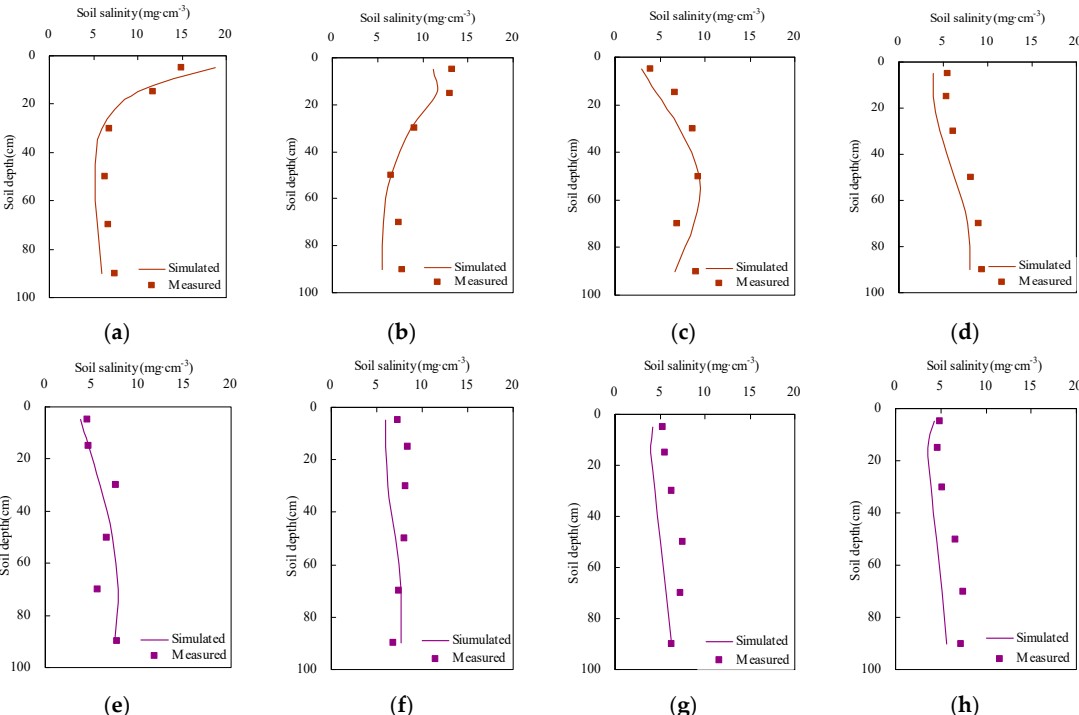

**Figure 6.** Comparison of the simulated and measured soil salinity at different times: (**a**) 28 June 2013; (**b**) 19 July 2013; (**c**) 31 July 2013; (**d**) 27 August 2013; (**e**) 11 June 2014; (**f**) 29 June 2014; (**g**) 24 July 2014; (**h**) 15 August 2014.

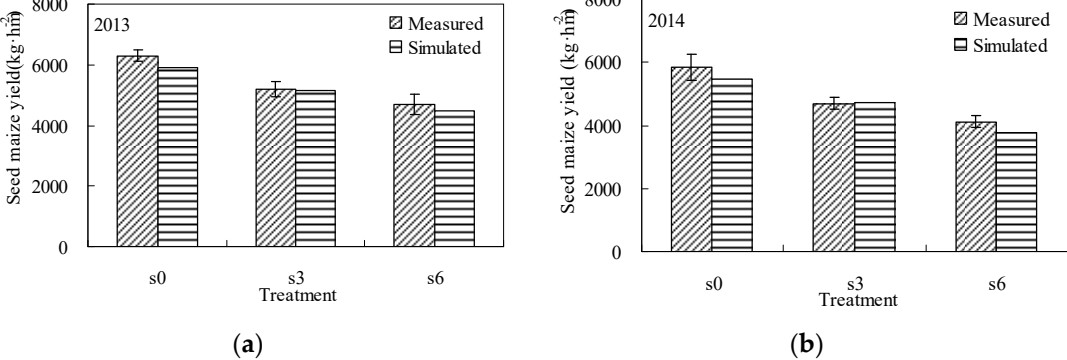

**Figure 7.** Comparison of the simulated and measured maize yield in calibration 2013 (**a**) and validation 2014 (**b**).

These results demonstrated that the calibrated and validated SWAP model was able to accurately simulate soil water–salt transport and maize yield, and predict these under long-term saline water irrigation with waters of different salinity in this study area.

*3.2. Simulation of Saline Water Irrigation with Different Water Salinity on Soil Water–Salt Transport and Maize Yield*

Because field experiments were restricted by various factors, it was difficult to comprehensively carry out saline water irrigation experiments with waters of different salinities. The calibrated SWAP model can be used to simulate saline water irrigation with different water salinity. The simulated irrigation water salinities tested were 0.71, 1.0, 2.0, 3.0, 4.0, 5.0, 6.0, 7.0, and 8.0 mg/cm$^3$. As the Shiyang River Basin is located in an arid desert area where rainfall is scarce, there is little difference between normal flow years, and dry or wet years. During the simulation, the meteorological data were based on the average annual method ($p$ = 50%). Drawing on 1960–2014 rainfall data, rainfall frequency

analysis showed that the typical year with $p = 50\%$ was 2011, when rainfall during the maize growth period was 118.8 mm. Based on local irrigation experience and previous studies, maize is generally irrigated four or five times over its growth period. Irrigating for the fifth time is mainly to maintain maize grain plumpness, thereby obtaining a greater yield and economic income. The added irrigation occurs in the mature stage of the maize crop cycle [25]. The present simulation did not consider a fifth irrigation, but rather irrigated four times during the maize growth period, according to the crop's water requirements. The overall irrigation water quota was 450 mm, including 120 mm at the seedling stage (5 June), 120 mm at the jointing stage (30 June), 105 mm at the heading stage (20 July), and 105 mm at the filling stage (10 August). The SWAP was simulated based on initial soil water content, soil salinity, and crop growth data of the s0 treatment in 2013. The simulated soil depth range was 0–100 cm. Under these conditions, the calibrated SWAP model was used to simulate saline water irrigation with nine different levels of irrigation water salinity. Figure 8a shows that, although the irrigation water amount was unchanged, irrigation water of different salinities altered soil water transport. The soil water content increased gradually with an increase in irrigation water salinity. At the higher irrigation water salinities, salt stress worsened, inhibiting root water absorption by the crop, and allowing more water to remain in the soil [26]. The soil water content under irrigation water salinities of 0.71, 1.0, and 2.0 mg/cm$^3$ was basically the same, indicating that soil water content was almost the same under low irrigation water salinities. Soil salinity in the 0–100-cm soil layer under saline water irrigation increased with an increase in irrigation water salinity (Figure 8b). At the end of the maize growth period, the soil salinity under irrigation with waters of 0.71, 1.0, and 2.0 mg/cm$^3$ salinities was basically the same; however, for waters with salinities of 3.0, 4.0, 5.0, 6.0, 7.0, and 8.0 mg/cm$^3$, it increased by 3.73, 5.03, 6.20, 7.23, 8.18, and 9.05 mg/cm$^3$, respectively, compared to the 0.71 mg/cm$^3$ irrigation water. Simulated soil water flux and salt flux changed in a similar manner in the 100-cm soil layer (Figure 9); the downward soil water flux and salt flux gradually increased with an increase in irrigation water salinity, reflecting the fact that soil water content could promote soil salinity movement to a certain degree.

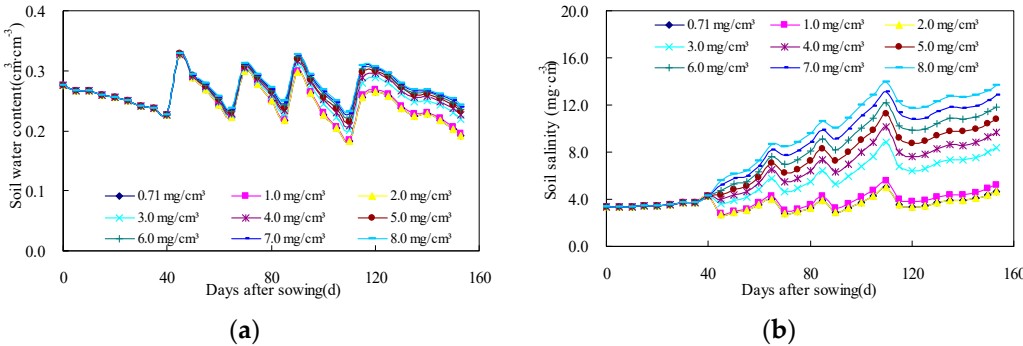

**Figure 8.** Simulation of soil water content (**a**) and soil salinity (**b**) with different saline water irrigation for the 0–100-cm soil layer.

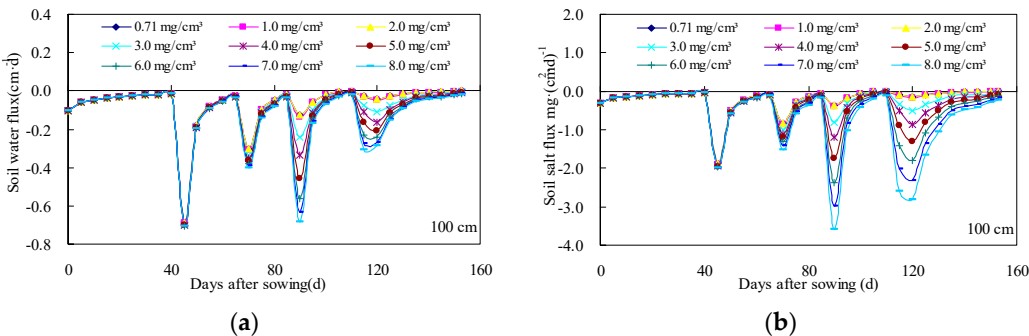

**Figure 9.** Simulation of soil water flux (**a**) and salt flux (**b**) with different saline water irrigation for the 100-cm soil layer.

Soil water–salt balance under saline water irrigation with different water salinities is presented in Table 5. A negative sign for soil water change indicates that soil water content was diminished through consumption. A negative sign for bottom soil water flux indicated that soil water was leaking downward. The bottom soil salt flux was negative, indicating that soil salinity moved downward. The increase of soil salinity was negative, indicating that soil salinity was leached. Based on the principle that a lesser increase in soil salinity would result in a greater yield and water use efficiency for maize, suitable irrigation water salinity was sought for the study area. It can be seen from Table 5 that the soil salinity of the 0.71, 1.0, and 2.0 $mg/cm^3$ irrigations increased below 50 $mg/cm^2$, the maize yield was above 5168 $kg/hm^2$, and the maize water use efficiency was above 1.0 $kg/m^3$. Therefore, brackish water of 0.71–2.0 $mg/cm^3$ can be used for irrigation in the study area. The soil salinity of 6.0, 7.0, and 8.0 $mg/cm^3$ irrigations increased more than 180 $mg/cm^2$. Compared with the simulated maize yield of 0.71 $mg/cm^3$, the maize yield decreased by more than 28% and the water use efficiency was less than 0.95 $kg/m^3$. Therefore, irrigation water salinity of 6.0, 7.0, and 8.0 $mg/cm^3$ was not suitable for irrigation. The soil salinity of 3.0, 4.0, and 5.0 $mg/cm^3$ irrigations increased between 80 and 160 $mg/cm^2$. The yield reduction of maize was between 12.5% and 23.0% and the water use efficiency was between 0.98 and 0.95 $kg/m^3$, which were slightly worse than brackish water of 0.71–2.0 $mg/cm^3$. Therefore, saline water of 3.0–5.0 $mg/cm^3$ can be used for irrigation over a short period of time. Whether saline water of 3.0–5.0 $mg/cm^3$ can be used for irrigation over a long period of time still needs further study.

### 3.3. Long-Term Simulation for Soil Salinity Transport and Maize Yield

Whether saline water of 3.0, 4.0, and 5.0 $mg/cm^3$ can be used for a long time in the study area was determined by allowing the SWAP model to predict the long-term effects of saline water irrigation bearing 3.0, 4.0, and 5.0 $mg/cm^3$ on soil salt transport and maize yield. In the simulation process, the initial soil water content, initial soil salinity, irrigation scheduling, and meteorological data were unchanged. Taking the simulation results of soil water content and soil salinity at the end of each year as the initial conditions for the next year, saline water irrigation with 3.0, 4.0, and 5.0 $mg/cm^3$ was continuously simulated for five years. The influence of irrigation water salinity on soil salinity transport in the 0–100-cm soil layer over the five-year period showed a year-to-year increase in soil salinity for all three irrigation water salinities (Figure 10a). The soil salinity increased sharply in the early stages of the simulation, and more slowly in the latter stages. At the end of the fifth year, the soil salinity of 3.0, 4.0, and 5.0 $mg/cm^3$ irrigations reached a steady state. After five years of simulation, the soil salinity of 3.0, 4.0, and 5.0 $mg/cm^3$ irrigations increased by 9.76, 10.95, and 11.94 $mg/cm^3$, respectively, compared with the initial soil salinity. Simulated maize yields over the five years indicated that maize yield declined with advancing years (Figure 10b). The maize yield decreased greatly in the early stages of simulation, and more slowly thereafter. After five years of simulation, maize yield reached a steady state. After five years of simulation, the maize yields under saline water irrigation with 3.0, 4.0, and 5.0 $mg/cm^3$ were 4097.18, 3718.98, and 3403.81 $kg/hm^2$, decreasing by 26.14%, 32.95%, and 38.64%, respectively, compared with the simulated yields under saline water irrigation with 0.71 $mg/cm^3$ in the first year. The simulated annual declines in maize yield for saline water irrigation at a salinity of 3.0 $mg/cm^3$ were 12.5%, 23.86%, 25.0%, 26.14%, and 26.14% from the first to the fifth year, respectively, compared with the simulated yield for maize in the first year for irrigation water bearing a salinity of 0.71 $mg/cm^3$. Over a long period of time, an irrigation water salinity of 3.0–5.0 $mg/cm^3$ would gradually increase soil salinity and have a significant negative impact on maize yield. The maize yields under saline water irrigation with 3.0–5.0 $mg/cm^3$ decreased by over 25% after five years. Therefore, saline water of 3.0–5.0 $mg/cm^3$ was not suitable for irrigation over a long period of time.

**Table 5.** Simulation of soil water and salt balance for different saline water irrigation.

| Irrigation Water Salinity (mg/cm$^3$) | Water Balance | | | | | Salt Balance | | | Yield (kg/hm$^2$) | Water Use Efficiency (kg/m$^3$) |
| --- | --- | --- | --- | --- | --- | --- | --- | --- | --- | --- |
| | Irrigation (mm) | Rainfall and Interception (mm) | Soil Water Change (mm) | Bottom Flux (mm) | Evapotranspiration (mm) | From Irrigation (mg/cm$^2$) | Bottom Flux (mg/cm$^2$) | Increase in Soil (mg/cm$^2$) | | |
| 0.71 | 450 | 108.7 | −107.8 | −118.1 | 548.4 | 31.95 | −35.95 | −4.00 | 5546.96 | 1.01 |
| 1.0 | 450 | 108.7 | −103.9 | −120.3 | 542.3 | 45.00 | −36.75 | 8.25 | 5483.92 | 1.01 |
| 2.0 | 450 | 108.7 | −91.0 | −129.9 | 519.8 | 90.00 | −40.67 | 49.33 | 5168.76 | 1.00 |
| 3.0 | 450 | 108.7 | −80.7 | −142.4 | 497.0 | 135.00 | −46.61 | 88.39 | 4853.59 | 0.98 |
| 4.0 | 450 | 108.7 | −73.0 | −157.1 | 474.6 | 180.00 | −55.19 | 124.81 | 4538.42 | 0.96 |
| 5.0 | 450 | 108.7 | −67.3 | −173.6 | 452.4 | 225.00 | −66.86 | 158.14 | 4286.28 | 0.95 |
| 6.0 | 450 | 108.7 | −62.8 | −190.7 | 430.8 | 270.00 | −81.49 | 188.51 | 3971.12 | 0.92 |
| 7.0 | 450 | 108.7 | −59.3 | −207.8 | 410.2 | 315.00 | −98.57 | 216.43 | 3718.98 | 0.90 |
| 8.0 | 450 | 108.7 | −56.4 | −224.4 | 390.7 | 360.00 | −117.80 | 242.20 | 3466.85 | 0.89 |

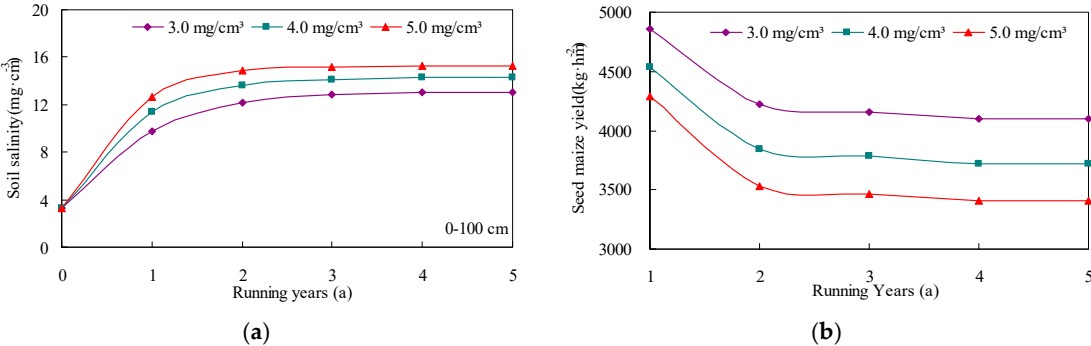

**Figure 10.** Simulation of soil salinity transport (**a**) and maize yield (**b**) over five years.

This study's results were consistent with those of Wu et al. [27], who found that many years' irrigation of winter wheat with water bearing a salinity of 3.0 mg/cm$^3$ in northern China's Huanghuai-Hai Plain led to salt accumulation in the upper soil layer. They concluded that, if saline water with a degree of mineralization of 3.0 g/L was utilized continuously, plants would be subjected to salt stress. However, the salt in the upper soil layer could be kept in the normal range for crop growth under sufficient rainfall and leaching. After five years of saline water irrigation with 3.0 mg/cm$^3$, the soil salinity increased slightly in the later stages and reached a steady state in this study. Crescimanno and Garofalo [28] used the SWAP model to explore different saline water irrigation scenarios, which aimed at optimizing irrigation in a Sicilian vineyard. They found that irrigation water salinities of 2.1 dS/m and 6.2 dS/m applied at a fixed annual volume of 112 mm proved to be the best strategy, reducing soil salinization and enhancing crop transpiration. This is similar to the finding that saline water of 3.0–5.0 mg/cm$^3$ could be used for irrigation in the short term in this study. However, the difference was that saline water of 3.0–5.0 mg/cm$^3$ was not suitable for irrigation in the long term in this study. Martínez et al. [29], simulating the water balance components in an irrigated citrus orchard in Andalucía (southern Spain) with the SWAP model, found it to be useful in supporting the development of appropriated irrigation strategies for sustainable water management in Andalucía. The model could be used as a simulation tool of saline water irrigation to simulate soil water–salt transport and maize yield in this study. Therefore, the SWAP model could be used to vet saline water irrigation strategies for sustainable water management in semi-arid and arid areas [30,31].

## 4. Conclusions

The RMSE and MRE values were less than 0.05 cm$^3$/cm$^3$ and 20%, respectively, for soil water content in the SWAP model calibration and validation. For soil salinity, the RMSE values were all lower than 5.0 mg/cm$^3$, and the MRE values were lower than 25%. The RMSE and MRE values were less than 380 kg/hm$^2$ and 10%, respectively, for maize yield. The simulated soil water content, soil salinity, and maize yield agreed well with the measured values. The SWAP model was able to simulate soil water–salt transport and maize yield for different saline water irrigations in the study area. The simulation results of saline water irrigation with different water salinities showed that brackish water of 0.71–2.0 mg/cm$^3$ can be used unrestrictedly for irrigation. The yield of maize was above 5168 kg/hm$^2$ and the water use efficiency of maize was above 1.0 kg/m$^3$. Irrigation water salinity above 6.0 mg/cm$^3$ was not suitable for irrigation. The yield of maize was reduced by above 28% and the water use efficiency of maize was below 0.95 kg/m$^3$. The use of irrigation water with a salinity of 3.0–5.0 mg/cm$^3$ could be acceptable for short-term irrigation as the yield of maize was reduced by 12.5–23.0% and the water use efficiency was 0.95–0.98 kg/m$^3$. Long-term simulation results of saline water irrigation with 3.0–5.0 mg/cm$^3$ indicated that soil salinity would gradually increase over the years. After five years, the soil salinity of 3.0–5.0 mg/cm$^3$ increased by above 9.5 mg/cm$^3$ compared with the initial soil salinity. At this irrigation water salinity, maize yield decreased over the years, dropping by over 25% after five years. An Irrigation water salinity of 3.0–5.0 mg/cm$^3$ was deemed

unsuitable for long-term irrigation in the study area. This study is important in terms of saline water irrigation and agricultural production practices in the Shiyang River Basin of northwest China.

**Author Contributions:** C.Y. and Q.J. collected the related data; C.Y. and Z.H. analyzed and built the SWAP model; C.Y. and S.F. analyzed the results and wrote the paper; Z.H. and S.F. modified this article.

**Funding:** This research was financially supported by the National Natural Science Foundation of China (51179166), Specialized Research Fund for the Doctoral Program of Higher Education of China (20123250110004), and a project funded by the Priority Academic Program Development of Jiangsu Higher Education Institutions (PAPD).

**Acknowledgments:** We are very grateful to Z.Q. (James H. Brace Associate Professor of McGill University, Canada) for his revision of the English language of our manuscript. We greatly appreciate to the editors and anonymous reviewers for their valuable comments.

**Conflicts of Interest:** The authors declare no conflicts of interest.

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
