# Peer review of "Simulation of Saline Water Irrigation for Seed Maize in Arid Northwest China Based on SWAP Model"

_sustainability, doi:10.3390/su11164264_

Round 1

Reviewer 1 Report

The authors present simulation of saline water irrigation for seed maize based on the SWAP model. It is an interesting study, of great practical importance, which can be used in areas with poor water resources. The created model was verified based on field research carried out in the arid northwest China.

Comments:
- chapter 2 lacks a map with the location of the research area and sample photographs illustrating the test plots.
- the authors do not provide data on daily precipitation during the experiment. They use only the average annual values from the period 1951-2013 or 2013 and 2014 years.
- the main disadvantage of this study is the lack of discussion in chapter 3. The authors did not compare the results obtained with similar studies carried out by other authors and in other dry areas (not only in China). This part of the study needs to be corrected.
- lack of information on the size of irrigated areas in China in arid areas and their impact on sustainable irrigation management of areas of dry suitability of the applied model
- conclusions should be more general, too many detail should be avoided

Technical notes:
- line 431 is "83: 22-29" and should be "83, 22-29"
- line 433 is "41: 36-43" and should be "43, 36-43"

Reviewer 2 Report

Dear Authors,

the paper verified the usefulness of the saline water to irrigate the seed maizein in tha arid Nothwest China. The results give insight about the usage of this irrigation tecniques where others water sources sortage are not present.

In the overall tha paper is well structured and is easly readble, but the english language may be improved by a native. The paper is accetable with minor revision. In the following   some few remarks:

A Figure on the geographical localization of the area is need, in the paragraph 2.1, with a detailed map where localize the experimental area, the meaurements stations and the weather station. 

pag 3 line 106 "used to irrigation" change in "used for irrigation"

pag.3 line 108 change "which" in "with" and add "of" before "25 cm"

pag. 4 line 138, delete "which"

pag 5 line 178, is "experiment station" modificable in "experiment area"?

Best regards

Reviewer 3 Report

In this manuscript, the SWAP model was calibrated and validated based on saline water irrigation experiments data. The model used to simulate soil water-salt transport and seed maize yield for different saline water irrigation. The study area is Arid Northwest China.

I found the subject of the manuscript interesting, but the English writing needs improvements. The paper will not be understandable by the international audience of the Sustainability Journal.

Furthermore, I found the abstract weak and authors need to rewrite abstract again in a more scientific way.

In Abstract, Authors repeat in continuous paragraphs the same words. Example 1: The SWAP model  was calibrated and validated based on field experiments data. The SWAP model was used to simulate soil water-salt transport and seed maize yield after calibration and validation.
It would better to just write in the 2nd paragraph.... The model was used to simulate soil water-salt transport and seed maize yield after calibration and validation.

Example 2: from...The SWAP model could be used as a simulation tool of saline water irrigation .....until... could be acceptable irrigation for a short time in the study area. In these 4 paragraphs authors repeat at the end of each paragraph the words study area.

The same in the Conclusions chapter: "The SWAP model" repeated in continius paragraphs. Since the manuscript is about The SWAP model it would be better to use similar expressions such: model or just like SWAP. The same with the words RSME and MSE.
